# Preliminary Assessment of the Possible Environmental Risks of Photopolymerizing Resin Particles Produced by Finishing Stereolithography 3D-Printed Objects, Employing Toxicity Test on Tropical House Crickets (*Gryllodes sigillatus*)

**DOI:** 10.3390/ijms262311245

**Published:** 2025-11-21

**Authors:** Bogumił Łosiewicz, Maciej Kamaszewski

**Affiliations:** 1Department of Genetic Engineering, The Kielanowski Institute of Animal Physiology and Nutrition, Polish Academy of Sciences, Instytucka 3, 05-110 Jabłonna, Poland; b.losiewicz@ifzz.pl; 2Department of Animal Environment Biology, Institute of Animal Sciences, Warsaw University of Life Sciences-SGGW, Ciszewskiego 8, 02-786 Warsaw, Poland

**Keywords:** additive manufacturing, biochemistry, ecotoxicology, Gryllidae, microparticles, nanoparticles, photopolymerizing resin

## Abstract

Additive manufacturing (AM), also known as 3D printing, is a rapidly growing field in industry. AM technologies include sintering, melting, and stereolithography. With steadily rising utilization, evaluating the environmental impact of AM materials has become essential, as these materials may act as emerging pollutants. Photopolymerizing resins (PRs) used in stereolithography can enter terrestrial ecosystems in polymerized and unpolymerized forms due to improper disposal. Insects are likely to be among the first organisms exposed to these contaminants in land ecosystems. This study evaluates the physiological effects of photopolymerizing resin particles (PRPs) produced via sanding on tropical house crickets (*Gryllodes sigillatus*) that were fed PRPs-contaminated agarose gels for 10 days. Effects were evaluated through mortality observations and enzymatic activity assays of cell transport mediating enzymes, digestive enzymes, and antioxidative stress enzymes. PRPs exposure caused sex-dependent differences in survival; an increase in amylase, alanine aminotransferase, aspartate aminotransferase, and trypsin; and a decrease in alkaline phosphatase, glutathione peroxidase, and superoxide dismutase activity, indicating molecular and cellular damage. PRPs’ toxicity might be enhanced due to a sex-dependent pulverization capability exhibited by *G. sigillatus*. These findings underscore the potential ecological risks associated with PRPs in terrestrial environments and the need for further research on their environmental impact.

## 1. Introduction

Additive manufacturing (AM), also known as 3D printing, is a relatively new and rapidly growing branch of the manufacturing industry that originated in the 1970s [1,2]. AM technologies are based on the three main types of printing: sintering, where, to compose prints, the printing material is heated without being liquified; melting, where electron beams are used to melt powders; and stereolithography (SLA), which utilizes photopolymerization, in which an ultraviolet laser in a photopolymer resin vat catalyzes the resin photopolymerization. AM technologies have been widely adopted across various industries, including medical implants, electronics, and aerospace [2]. With the increasing utilization of AM techniques, particularly those involving photopolymer resins, there is a growing need to evaluate the environmental impact of both the manufacturing process and the raw materials involved [3,4,5].

Evaluating the environmental risks associated with AM techniques has become especially important in recent years as AM has undergone rapid technological advancement [5,6,7,8]. Due to these advancements, stereolithography has emerged as a widespread technique among hobbyists. The use of photopolymerizing resins means that SLA enables the production of high-resolution printed objects characterized by fine structural detail and smooth surface finishes, which are especially valued in a number of applications, such as the fabrication of detailed figurines. Recent innovations—including faster printing speeds; enhanced resin formulations; and the availability of cost-effective, desktop-scale SLA printers—have significantly lowered entry barriers. As a result, SLA has gained widespread adoption among hobbyists and small-scale creators. The convergence of affordability, intuitive software interfaces, and the capacity to fabricate intricate components has positioned SLA as a favored method for prototyping, artistic creation, and custom manufacturing in non-industrial contexts [9]. However, the growing use and accessibility of SLA technologies have raised environmental concerns, particularly regarding the improper handling and disposal of photopolymerizing resin (PR) waste. These practices can lead to the release of both polymerized and unpolymerized resin particles into the environment [5,10,11]. Such contamination is especially concerning in terrestrial ecosystems, where invertebrates may inadvertently ingest these resin particles, posing risks to their health and to the broader ecological balance.

Current toxicity research on PRs has primarily focused on the toxicity of uncured resins [12], their biocompatibility in biomedical applications [3,13,14,15], and the potential risks associated with exposure to airborne particles [16,17]. The residues of unpolymerized resin have been detected in freshly printed objects, raising concerns about the safety of medical implants fabricated using these materials and highlighting the necessity for post-processing treatments [13,14,15,18,19]. Despite the existing research, significant knowledge gaps remain concerning the impact of the PRs used in 3D printing on both living organisms and the environment. Most notably, in animal models, the effects of photopolymerizing resin particles (PRPs) have not been well studied [4]. Regardless of whether the particles are released from printed objects or post-printing waste, there is a lack of studies investigating the toxicity of PRPs. Currently, published research has focused on the effects of printed objects and their leachates on *Caenorhabditis elegans* [20], *Daphnia magna*, and zebrafish (*Danio rerio*) [14,19,21] and investigated the toxicity of unpolymerized resin on *Ceriodaphnia dubia* [12]. Research using animal models shows that PRs’ leachates cause mortality and behavioral changes and disrupt growth. The latest published research further investigated the possibility of improving the biocompatibility of SLA-printed objects and verifying biocompatibility on bacteria and cell line models [22]. Concerns about the impact of SLA materials on the environment are expected to become increasingly relevant as the use of these materials expands, particularly given the risks associated with inadequate waste disposal and unintended exposure to fabricated models. To address this issue and the lack of knowledge about the effects of PRPs on terrestrial animals, we conducted a preliminary, 10-day study on the oral toxicity of polymerized photopolymerizing resin particles on the tropical house cricket (*Gryllodes sigillatus*) model. This is a known model species and has been employed in numerous studies over many years [23,24,25,26,27,28]. Moreover, various cricket species are increasingly utilized as model organisms in nutritional [29,30,31] and toxicological research [32,33,34]. *G. sigillatus* is an omnivorous species native to the tropical and subtropical regions of Asia and has been introduced to various other geographic regions such as Japan and Mexico [35,36,37]. Given the increasing relevance of insects in both animal and human nutrition, *G. sigillatus* has emerged as a species of particular interest. This is primarily due to its rapid growth rate, high reproductive capacity, and favorable palatability. Additionally, *G. sigillatus* is among the limited number of insect species that have been authorized for commercial farming within the European Union, further underscoring its suitability for large-scale rearing [37,38,39,40]. Therefore, considering the potential of the tropical house cricket as a model organism, this study aimed to conduct research on the effects of PRPs on the survival and physiology of a terrestrial arthropod capable of synanthropy. The findings of this study may inform future research about the possible mode of action of PRPs in crickets and thus lead to better experiment designs. Moreover, the results that are described below may provide a foundation for environmental preservation policies concerning 3D printing waste management.

## 2. Results

### 2.1. Characteristics of Photopolymerizing Resin Particles

Using dynamic light scattering (DLS) analysis, the average measured hydrodynamic diameter of the PRPs ranged from 220 nm to 459 nm. The obtained mean zeta potential of the PRPs was −16.2 ± 11.6 mV (Figure 1A,B).

In the dynamic nanoparticle tracking analysis (NTA), the mean diameter of the obtained PRPs was 328 ± 213 nm, with a median of 232.5 nm, while 4.25%, 49.83%, and 45.92% were in the range of 0–100 nm, 100–300 nm, and 300–1000 nm, respectively. In the static NTA, the mean diameter of the obtained PRPs was 320 ± 133 nm, with a median of 248.5 nm, while 2.70%, 49.35%, and 47.95% were in the range of 0–100 nm, 100–300 nm, and 300–1000 nm, respectively (Figure 1C).

The measurements of the PRPs from transmission electron microscopy (TEM) images ranged from 31.4 nm to 752.6 nm, with a median of 178.3 nm and a mean of 200.9 ± 97.7 nm for particle diameters, while agglomerate diameters were in the range of 64.6–8575 nm, with a median of 250.9 nm and a mean of 813.3 ± 1493.5 nm. One of the images obtained is presented in Figure 2.

### 2.2. Mortality and Observations Made During the Toxicity Test

The body measurements of the crickets (*n* = 150) were homogeneous across all experimental groups and included the following parameters: length 15.39 ± 1.57 mm and thorax width 3.90 ± 0.44 mm. Due to the crickets being divided into groups, instances of necrophagy were observed throughout the experiment, occurring frequently in all PRPs-exposed groups and twice within one of the replicates of the control group; this was evidenced by visible signs of tissue consumption on deceased individuals. However, no aggressive cannibalistic behavior was detected; notably, no additional mortality was recorded on the day following molting in those containers where molting had occurred. Mortality increased progressively with increasing duration of exposure. The highest mortality rate was observed in the 2000 mg/dm^3^ group, followed by the 200 mg/dm^3^, control, and 20 mg/dm^3^ groups, while the lowest was recorded in the 2 mg/dm^3^ group (Figure 3). There were no statistically significant differences between the groups in terms of mortality rate, according to the Kruskal–Wallis test, *p* > 0.05. Observed mortality was sex-dependent according to the results of the χ^2^ test: χ^2^ (dF = 1, *n* = 119) = 20.0, *p* < 0.01. The distribution of mortality grouped by sex ranged from 43% to 100% for males and from 0% to 57% for females for all deaths, with the closest group to a 50:50 ratio being the control group (Table 1).

### 2.3. Biochemical Assay Results

The enzymatic activity of alkaline phosphatase (ALP), alanine aminotransferase (ALT), asparagine aminotransferase (AST), amylase, glutathione peroxidase (GPX), superoxide dismutase (SOD), and trypsin showed statistically significant differences among the experimental groups. No significant differences were observed in acid phosphatase (ACP) and lipase activity. ACP activity did not vary significantly among groups; however, the lowest values were found in the control and 20 mg/dm^3^ groups, while the highest activity was recorded in the 200 mg/dm^3^ group. Contrary dependency was observed in lipase activity and was lowest in the 2 mg/dm^3^ and 200 mg/dm^3^ groups and highest in the control, 20 mg/dm^3^, and 2000 mg/dm^3^ groups. Comparable results were obtained for the trypsin activity assays, in which the highest enzyme activity was observed in the 2000 mg/dm^3^ group and the lowest in the 2 mg/dm^3^ group. ALP activity was highest in the control group, while all groups exposed to PRPs exhibited reduced activity, with the lowest level observed in the 2000 mg/dm^3^ group. A similar trend was noted for amylase activity: the control group displayed the highest activity, the 200 mg/dm^3^ group showed the lowest, and no statistically significant differences were found among the other PRPs-exposed groups (Figure 4).

In contrast to the observed results for the ALP and amylase assays, the ALT and AST activities showed an inverse pattern. The control group exhibited the lowest activity, while the highest levels were recorded in the 200 mg/dm^3^ group. Intermediate and statistically comparable ALT activities were observed in the 2 mg/dm^3^, 20 mg/dm^3^, and 2000 mg/dm^3^ exposure groups. GPX activity exhibited a concentration-dependent decrease, with the highest activity being in the control group and the lowest in the 2000 mg/dm^3^ group. SOD activity followed a similar pattern, with the highest values in the control group and the lowest in the 20 mg/dm^3^ group. A slightly higher SOD activity was noted in the 200 mg/dm^3^ and 2000 mg/dm^3^ groups (Figure 5).

## 3. Discussion

In recent years, the tropical house cricket has emerged as a new, viable model for toxicology research, especially for assessing the effects of microplastics [25,27,41]. Moreover, in the future, they could be employed in multigenerational toxicology studies to assess the impact of xenobiotics on fecundity due to their relatively short lifespan of 50–70 days. The lifespan of *Gryllodes sigillatus* depends on whether the crickets are from an inbred or outbred line, and on their sex; thus, the mean life span is 58.45 ± 0.94 days for males and 43.70 ± 0.60 days for females [42]. Additionally, discoveries could be made about how emerging pollutants affect these crickets and whether they accumulate xenobiotics. These discoveries could be valuable for environmental impact assessments and the evaluation of the possible biotransfer of xenobiotics to the predators of crickets [43].

Interestingly, the lowest observed mortality data was not in the control group. Notably, higher mortality was observed in the control group compared to the 2 mg/dm^3^ treatment group. Similar findings were presented by Ritchie et al. (2024) [25] on one-week-old tropical house crickets. Also, comparable results were reported by Fudlosid et al. (2022) [27], who observed equal or lower mortality in groups exposed to low concentrations of microplastics compared to controls. These findings highlight the need to further study how low concentrations of synthetic polymers affect *G. sigillatus*, as they might act as fiber and ease the passage of gut contents. The obtained standard deviations for mortality might have been influenced by the relatively small number of specimens per replicate group. The occurrence of necrophagy in all the exposed groups suggests that tropical house crickets may possess a similar mechanism to ants, fruit flies (*Drosophila melanogaster*), and black fungus gnats (*Bradysia difformis*): the avoidance of micro- and nano-plastics, and phthalates in feed [44,45,46,47,48]. This mechanism is not present in all insects, as honeybees do not show a preference for uncontaminated food [49]. However, the latest research by Ritchie et al. (2025) [43] indicates that *G. sigillatus* may even develop a slight preference for microplastic-contaminated food. The observation of rare instances of necrophagy in the control group suggests that natural scavenging behavior on dead insects does occur in these omnivorous species, which might also have happened in the exposed groups. Notably, necrophagy has also been reported in the closely related species *Gryllus assimilis* [37]. Thus, in the future, food preference should be tested to verify whether tropical house crickets exhibit a mechanism for avoiding food contaminated with PRPs, or whether they can develop a slight preference for it, as they did for polyethylene microparticles, as reported by Ritchie et al. (2025) [43]. To improve feeding conditions and maintenance, separate trays for water and gel might be used; however, the minimal volume of water that was deposited on the gel prevented them from becoming desiccated. A proposed model for a tray divided into two parts is available in Appendix A, although the use of trays for feed and test tubes for water, as is often utilized, is also a viable option [26,30].

The observed mortality is consistent with previous studies investigating the toxicity of polymerized PRs, which have demonstrated toxic effects on cell lines and zebrafish (*Danio rerio*), primarily due to leachates from incompletely cured, freshly printed materials [13,14,15,19]. The toxicity of uncured photopolymer resin has also been confirmed in *Ceriodaphnia dubia* [12]. Furthermore, findings by Camassa et al. (2021) [16] indicate that dental resin dust can induce cytotoxic responses in humans’ bronchial epithelial cells (HBEC-3KT) after exposures exceeding 24 h. In the present study, a similar delayed toxicity was observed, as mortality increased notably between days three and five in all exposed groups. A continuous rise in mortality was particularly evident in the 200 mg/dm^3^ group from day three onward. These findings align with previous reports on the effects of micro- and nano-plastics on the Japanese carpenter ant (*Camponotus japonicus*) and silkworms (*Bombyx mori*) [48,50]. Enzymatic activity profiles further support evidence of toxicity. A slight increase in ACP activity suggests cytotoxic effects, while a decrease in ALP activity among the exposed groups, compared to the control, indicates potential damage to intestinal epithelial cells, as ALP is a brush border enzyme abundantly expressed in these tissues [38]. This extends to ALT and AST activity, which increased in all exposed groups, thus further suggesting digestive tract damage as both enzymes may serve as an indicator of cellular damage in both vertebrates and invertebrates [38,51]. This is consistent with results obtained by Inagaki et al. (2012) [52], as cytotoxic drugs injected into silkworms increased ALT activity in hemolymph, and in accordance with data obtained by Mirhaghparast et al. (2015) [53], where hexaflumuron caused an elevation in the hemolymph ALT and AST activities of *Chilo suppressalis* over time. Digestive enzyme activities (amylase, lipase, trypsin) show a similar reaction to PRPs, as demonstrated in a study conducted on silkworms by Muhammad et al. (2022) [54] in which amylase activity was inhibited by copper oxide (CuO) and zinc oxide (ZnO) nanoparticles. In their study, there was no statistically significant impact of CuO and ZnO nanoparticles on lipase activity, while trypsin was stimulated by ZnO nanoparticles but inhibited by CuO nanoparticles. However, our findings on digestive enzymes activities contradict results obtained by Bharani and Namasivayam (2017) [55], in which biosynthesized silver nanoparticles strongly inhibited digestive enzymes. The observed changes in digestive enzyme activity may also have arisen due to alterations associated with the introduction of a novel diet for the toxicological assay, which could be a topic for further studies. Certain individuals may require an extended period to adapt to altered trophic conditions, which could have affected enzymatic activity. Enzymes associated with the antioxidant response observed in this study also diverge from previous studies on plastic microparticles, which typically stimulated SOD and GPX activity. Instead, the antioxidant enzyme reaction observed in this study is comparable to the effects of plastic and metal-oxide nanoparticles on insects, in which these enzymes are inhibited [50,56,57,58,59]. This reflects the known capacity of nanoparticles to induce oxidative stress [60,61], although the response appears to vary by habitat. In aquatic species, SOD and GPX are stimulated by exposure to nanoparticles [62,63], while in terrestrial species, these enzymes are inhibited and accompanied by upregulation of catalase and glutathione-S-transferase [50,56,57,58,59]. Furthermore, all the obtained enzyme activity results comply with the nonlinear reaction of enzyme activity to nanoxenobiotics [56,57,59]. These results would not be surprising if, in the study, photopolymerizing resin nanoparticles were used instead of PRPs, which were mostly in the microparticle range. This might be connected to a mechanism observed by Ritchie et al. (2024) [25] in which tropical house crickets fed with polyethylene (PE) microplastics broke down PE beads during the early digestive process to a near nanoparticle size scale. The described pulverization process varies slightly depending on the specimen’s sex. A similar observation was made by Helmberger et al. (2022) [64] in a study where tropical house crickets fed with polyethylene foam fragmented it to a microparticle level. However, due to methodological limitations, the authors were unable to characterize the process. Thus, this ability to break down synthetic particles might have occurred in this study, resulting in the early breakdown of PRPs into smaller particles, and might have led to a mode of PRP action closer to nanoparticles, despite nanoparticles being a small fraction of the particle solution. Furthermore, the observed sex-related differences in mortality might be related to males’ more effective ability to break down particles into smaller sizes compared to females [25]. However, observed mortality differences might have been slightly affected by the uneven sex distribution in groups and should be further studied in the future, along with the possible occurrence of a pulverization mechanism.

## 4. Materials and Methods

### 4.1. Ethical Statement

The experiment did not require the consent of the local ethics committee, in accordance with the Polish Law on the Protection of Animals Used for Scientific or Educational Purposes.

### 4.2. Acquisition of Photopolymerizing Resin Particles

To obtain the PRPs, a process simulating the surface finishing of 3D-printed models was employed. Hollow resin blocks measuring 30 × 30 × 30 mm with a 20 × 20 × 25 mm hollow were fabricated using a photopolymer resin (Build, Siraya Tech, Hacienda Heights, CA, USA) via stereolithography carried out on a Phrozen Sonic Mega 8K V2 printer (Phrozen Tech Co., Ltd., Hsinchu City, Taiwan). The printing parameters were set in accordance with the manufacturer’s specifications. The printed blocks were subsequently grated using damp, waterproof sandpaper (grit: P180) (Starcke GmbH & Co. KG, Melle, Germany) to obtain an aqueous suspension containing resin particles. This suspension was then transferred to plastic microbial culture dishes and dried at 37 °C for 24 h to evaporate the water, yielding a dry resin dust suitable for subsequent analysis. During the sanding process, a minimal amount of force was applied to sand the blocks to ensure process repeatability.

### 4.3. Characterization of Particles

The obtained PRPs were suspended in deionized water at 20 mg/dm^3^ and 200 mg/dm^3^ concentrations for size characterization. They were then subjected to sonication lasting 30 min using an ultrasonic cleaner Ultron U-505 ultrasonic cleaner (Ultron, Dywity, Poland) at a frequency 40 kHz at 20 °C. The sonicated particles were subjected to characterization in aqueous solutions using dynamic light scattering analysis, transmission electron microscopy (concentration 20 mg/dm^3^), and nanoparticle tracking analysis (NTA) (concentration 200 mg/dm^3^). A DLS analysis was performed to determine the average hydrodynamic diameter and zeta potential of the obtained PRPs. Both analyses were conducted in triplicate using a Zetasizer Nano S90 analyzer (Malvern Instruments Ltd., Malvern, Worcestershire, UK). Observations of particles were made using a FEI 268D “Morgagni” transmission electron microscope (FEI Company, Hillsboro, OR, USA) equipped with an Olympus-SIS “Morada” digital camera (Olympus, Münster, Germany) and analyzed in QuPath software (v. 0.5.1) [65]. NTA measurements were performed on a NanoSight Pro analyzer (Malvern Panalytical, Malvern, Worcestershire, UK). The samples were injected using sterile syringes through a NanoSight syringe pump (Malvern Panalytical Ltd., Malvern, Worcestershire, UK) for both static and dynamic measurements. This provided a continuous flow of particles (flow rate 3 μL/min) into the sample chamber during the dynamic measurement. Measurements were performed at 25 °C at a wavelength of 488 nm. NanoSight Pro v1.2.0.3 software (Malvern Panalytical Ltd., Malvern, Worcestershire, UK) was used for data analysis.

### 4.4. Selection of the Tested Xenobiotic Concentrations

The PRP concentrations used in this experiment were selected based on the concentrations of microplastics used in experiments conducted by Fudlosid et al. (2022) [27], Ritchie et al. (2023) [41] and Ritchie et al. (2024) [25]. Based on these studies, in which *G. sigillatus* specimens were exposed to 0%, 0.25%, 0.5%, and 1%, and 0%, 2.5%, 5%, and 10% weight-to-weight ratios of microplastics in the feed, lower concentrations of PRPs were selected to account for their potentially higher toxicity. Therefore, 0 mg/dm^3^ (control), 2 mg/dm^3^, 20 mg/dm^3^, 200 mg/dm^3^, and 2000 mg/dm^3^ concentrations were used in the present study on tropical house crickets.

### 4.5. Agarose Gel Preparation

In this experiment, oral exposure to the xenobiotic was employed by incorporating it into agarose-based gels prepared as feed for *Gryllodes sigillatus*. Briefly, 9 g of sugar (Krajowa Grupa Spożywcza S.A., Toruń, Poland), 0.675 g of agarose (Merck Millipore, Darmstadt, Germany), and 4.5 g of Ichtiovit^®^ (Tropical, Chorzów, Poland) were weighed per group using a Radwag AS 82/220.R2 plus analytical balance (Radwag, Radom, Poland). The weighed components were then suspended in 45 mL of water solutions containing 0 mg/dm^3^, 2 mg/dm^3^, 20 mg/dm^3^, 200 mg/dm^3^, or 2000 mg/dm^3^ of resin particles. These solutions were composed of resin particles suspended in a prepared water solution containing 5.0 mM NaCl, 0.17 mM KCl, 0.33 mM CaCl_2,_ and 0.33 mM MgSO_4_. The resulting mixtures were heated in a microwave until complete dissolution of the sugar and agarose occurred, after which they were poured into molds. Once solidified, the gels were tightly sealed in aluminum foil and stored at 4 °C until use.

### 4.6. Scheme of the Experiment

Nymphs of the tropical house crickets were obtained from a commercial insect farm (Terr-Gal, Warsaw, Poland). The crickets were maintained at 23 ± 1 °C and 50 ± 10% humidity on a 10:14 L:D cycle in the frustum of rectangular-based, pyramid-shaped plastic boxes with the following dimensions: base, 75 × 5 mm; height, 105 mm; and top, 75 × 108 mm. In each box, a piece of egg carton was provided for shelter, along with food and water ad libitum. The base diet consisted of oatmeal and an apple, which also served as an additional water source. Food and water were replaced every 2–3 days till the end of the toxicology experiment.

The number of animals per group was determined based on a priori power analysis. Assuming a significance level of *α* = 0.05, a statistical power of 80%, and an estimated standard deviation of 1.2, we calculated that a minimum of 30 animals per group was required to detect a difference of approximately 0.87 units between groups.

A total of 150 tropical house crickets were used in the experiment. At the 4th instar, before the beginning of the experiments, individuals, without visible sex traits [66], were randomly assigned to five experimental groups: one control group and four treatment groups exposed to different concentrations of PRPs (2 mg/dm^3^; 20 mg/dm^3^; 200 mg/dm^3^; and 2000 mg/dm^3^). Each group had three replicates, with 10 individuals per replicate. At the start of the experiment, all specimens were photographed on millimeter paper to facilitate morphometric analysis, while sex distribution was also checked, which was 7:23; 13:17; 14:16; 19:11; 13:17 (male:female) for the groups 0 mg/dm^3^, 2 mg/dm^3^, 20 mg/dm^3^, 200 mg/dm^3^, 2000 mg/dm^3^ of PRPs, respectively. Subsequently, each replicate was housed in a separate container equipped with a tray for the gel and water, as well as a cut paper tube to provide shelter. Water and gel were provided ad libitum and replenished regularly: water was changed daily, while the gel was replaced every 2–3 days. Environmental conditions were monitored and maintained throughout the experiment, with a constant temperature of 21 ± 1 °C, humidity 50 ± 10%, and a photoperiod of 10 h of light and 14 h of darkness (L:D 10:14). The experiment was conducted over 10 days. On the final day, the gel was removed from all containers, and the crickets were subjected to 24 h of starvation before sampling. Crickets were observed daily to note the mortality rate and the possible occurrence of necrophagy, cannibalism, and molting, followed by the removal of decayed specimens.

### 4.7. Morphometric Measurements

Morphometric measurements, including total body length and thorax width, were conducted using QuPath software (v. 0.5.1) [65], based on the initial photographs. Additionally, sex and developmental stage were determined based on morphometric criteria as described by Kong et al. (2025) [66]. Each specimen’s sex was also determined during material sampling.

### 4.8. Mortality Calculations

Mortality rates were calculated for each treatment group with replicate subdivision. The mortality rate with sex division and the daily mortality dynamics were calculated without accounting for replicate subdivisions. Due to one individual escaping from the third replicate of the 2000 mg/dm^3^ group on the fifth day of the experiment, final mortality for this group was calculated based on 29 individuals instead of 30.

### 4.9. Enzyme Activity Assays

The crickets were euthanized by placing them in a glass beaker with a folded paper towel that had been soaked in acetone beforehand. Subsequently, 11, 13, 11, 11, and 9 specimens in the 0 mg/dm^3^, 2 mg/dm^3^, 20 mg/dm^3^, 200 mg/dm^3^, and 2000 mg/dm^3^ groups, respectively, were dissected into smaller pieces and transferred individually to 2 mL test tubes. The collected material was flash-frozen in liquid nitrogen and stored at −80 °C until further analysis. The samples’ homogenization was performed on ice in 500 µL of 0.01 M phosphate-buffered saline (PBS), pH 7.4 (Sigma Aldrich, St. Louis, MO, USA), at 4 °C. Then, the homogenates were centrifuged for 15 min at 14,000× *g* at 4 °C. The supernatants were collected, and the total protein concentration was quantified. Samples were then diluted in triplicate to achieve uniform protein concentrations, refrozen, and stored at −80 °C. The activity of ACP, ALP, ALT, AST, amylase, lipase, GPX, SOD, and trypsin was measured. Biochemical assays for total protein and the activity of ACP, ALP, ALT, AST, amylase, and lipase were conducted using Spinreact kits (Girona, Spain; REF 1001291, 1001122, 1001131, 41282, 41272, 41202, and 1001274, respectively). Activity measurements for GPX and SOD were performed using Randox kits (Randox Laboratories, Crumlin, UK; REF RS505 and SD125, respectively). All biochemical measurements were performed in 96-well plates at 37 °C, in triplicate, using a Tecan Infinite 200 PRO plate spectrometer (Tecan Austria, Grödig, Austria). Measurements for total protein content and the activity of ACP, ALP, ALT, AST, GPX, SOD, amylase, and lipase were performed according to the manufacturers’ kit methodologies which were adjusted to 96-well plates, as described by Kamaszewski et al. (2023) [63], Łosiewicz and Szudrowicz (2024) [51], and Szczepański et al. (2025) [67]. The detailed protocols for the biochemical assays and the composition of the reagents used are provided in Appendix A.

Total protein concentration and lipase activity were calculated according to the formula provided by the kit manufacturer. To calculate the activity of ACP, ALP, ALT, AST, GPX, and amylase in the sample, the mean delta absorbance between readings were substituted into the following formula:$$\left(△\frac{sample\;absorbance}{\min}\right)\times Factor=enzyme\;activity\;\left(\frac{U}{L}\right)$$

The mean delta absorbance was calculated by substituting the obtained absorbance readings into the following formulas (A—for ACP, ALP; B—for ALT, AST; C—for GPX; D—for SOD):$$A.\;△\frac{sample\;absorbance}{\min}=\frac{\left(A_{2}-A_{1}\right)+\left(A_{3}-A_{2}\right)+\left(A_{4}-A_{3}\right)}{3}$$$$B.\;△\frac{sample\;absorbance}{\min}=\frac{\left(A_{1}-A_{2}\right)+\left(A_{2}-A_{3}\right)+\left(A_{3}-A_{4}\right)}{3}$$$$C.\;△\frac{sample\;absorbance}{\min}=\frac{\left(A_{1}-A_{2}\right)+\left(A_{3}-A_{2}\right)}{2}$$$$D.\;△\frac{sample,\;calibrator,\;or\;blank\;absorbance}{\min}=\frac{\left(A_{2}-A_{1}\right)+\left(A_{3}-A_{2}\right)+\left(A_{4}-A_{3}\right)}{3}$$

A_1, 2, 3, 4_—measured individual absorbance values for samples/calibrator.

To find the coefficient for multiplying the obtained mean delta absorbance, the following formula was used:$$Factor=\frac{TV\times 1000}{AFoP\times SV\times P}$$

*TV*—total volume of the reaction mixture in the well (in mL)

*SV*—sample volume used for the reaction in the well (in mL)

*P*—optical path length

*AFoP*—absorbance factor of the product obtained in the reaction catalyzed by the measured product

*AFoP* used in the study:

14.7—absorbance factor for Azo Dye (at the 405 nm wavelength, measured in the ACP activity assay)

18.5—absorbance factor for p-Nitrophenol (at the 405 nm wavelength, measured in the ALP activity assay)

6.3—absorbance factor for dihydronicotinamide adenine dinucleotide and nicotinamide adenine dinucleotide phosphate (NADH, NADPH; at the 340 nm wavelength, measured in the ALT, AST, and GPX activity assays)

18.9—absorbance factor for 2-chloro-4-nitrophenol (CNP); at the 405 nm wavelength, measured in the amylase activity assay)

The optical path length was calculated using the following formula for flat-bottom 96-well plates:$$P=\frac{TV}{area\;of\;well\;bottom}$$

*TV*—total volume of the reaction mixture in the well (in cm^3^)

*P*—optical path length (height of the reaction mixture)

The area of the well bottom of the 96-well flat-bottom plates utilized in this study was 0.32 cm^2^, according to the manufacturer’s data sheet (NEST Biotechnology Co., Ltd., Wuxi, China). We do not recommend using other types of 96-well plates (e.g., V-bottom or U-bottom) for measurements that rely on optical path length in calculations, as the inability to accurately determine the optical path length may introduce calculation errors, leading to measurement inaccuracies.

To calculate the SOD activity in the sample, the change in △ absorbance/min for both the sample and the calibrators must be converted into percentage inhibition using the following equation:$$100-\frac{\left(△\frac{sample\;absorbance}{\min}\;or\;△\frac{calibator\;absorbance}{\min}\times 100\right)}{△\frac{blank\;absorbance}{\min}}=\%inhibition$$

The blank serves as an uninhibited reaction calibrator. The percentage inhibition values obtained for the calibrators should be plotted against the log10 of the concentrations of the calibrators and the blank in terms of SOD activity. The percentage inhibition of the sample can then be used to determine the SOD activity from the standard curve derived from the plotted data.

Trypsin activity was measured based on the formation of para-nitroaniline. Briefly, 45 mL of 0.05 M Tris-HCl (tromethamine hydrochloride) buffer at pH 7.5 was prepared. Then, on a 96-well plate, 10 µL of the samples, in triplicate, and 10 µL of 0.001267 µM para-nitroaniline for the standard curve (final standard concentration: 0%, 25%, 50%, 75%, 100%), in duplicate, were added. Afterwards, before performing the assay, 4.4 mg of BAPNA (Nα-benzoyl-DL-arginine-p-nitroanilide) (Sigma Aldrich, St. Louis, MO, USA) was dissolved in 100 µL of DMSO (dimethyl sulfoxide) (Sigma Aldrich, St. Louis, MO, USA) and topped up to 10 mL with deionized water. Then the Tris-HCl buffer and the substrate solution were thoroughly mixed, and 200 µL of the obtained solution was added to a previously prepared 96-well plate. The plate was gently shaken and incubated at 37 °C for 20 s. After the incubation, the initial absorbance was measured at 405 nm. Subsequently, the reaction was allowed to proceed for an additional 3 min, with absorbance readings taken at 1 min intervals. All obtained enzymatic activity measurements were standardized using total protein content.

### 4.10. Statistical Analysis

All quantitative data (body measurements, enzyme activities, and mortality rates without division for sex) were tested for normal distribution using the Shapiro–Wilk test. Subsequently, depending on the result (*p*-value higher or lower than 0.05), each parameter was analyzed using the appropriate statistical tests. Parameters with a normal distribution (body measurements, activity of ACP, ALT, AST, and SOD) were checked with one-way ANOVA followed by a post hoc Tukey test for varying n. Parameters with a non-normal distribution (activity of ALP, GPX, amylase, lipase, trypsin, and mortality rates without division for sex) were analyzed using the Kruskal–Wallis test to determine statistically significant differences. The association between mortality and sex was assessed using the chi-square (χ^2^) test. The mortality rate for each sex used to carry out the χ^2^ test was the total mortality calculated from all exposed groups, which resulted in the following data: 27 males and 6 females deceased and 31 males and 55 females surviving. All analyses were performed using STATISTICA (v13.3) software (Tibco Software, Palo Alto, CA, USA).

## 5. Conclusions

The toxicological assessments conducted on the tropical house cricket *Gryllodes sigillatus* yielded promising results for the future application of this species in toxicity testing; however, further optimization of the methodology is required. This study represents the first investigation into the physiological effects of PRP exposure for this terrestrial model organism. Toxicity slightly increased without statistically significant differences, which were visible in some of the biochemical assays in the 2000 mg/dm^3^ group, as evidenced by elevated mortality rates and alterations to enzymatic activity profiles. Exposure to PRPs primarily disrupted the activity of ALP, ALT, AST, amylase, trypsin, and the oxidative-stress-related enzymes, GPX and SOD. These enzymatic alterations suggest cellular damage resulting from PRP exposure. Notably, the enzymatic responses induced by PRPs that were mostly in the microparticle size range were comparable to those triggered by nanoparticles, which might be attributed to the sex-dependent pulverization capability exhibited by *G. sigillatus*. This mechanism may underlie the higher mortality observed in males relative to females; however, it should be further studied in future research to confirm. Additionally, the obtained results underscore the need for ongoing, multigenerational, and multitrophic level studies on the toxicity of PRPs. They may also initiate the development of waste management policies to mitigate possible environmental risks. Overall, the findings highlight the potential ecological risks associated with the environmental release of PRPs and support the use of *G. sigillatus* as a model organism in terrestrial ecotoxicology.

## Figures and Tables

**Figure 1 ijms-26-11245-f001:**
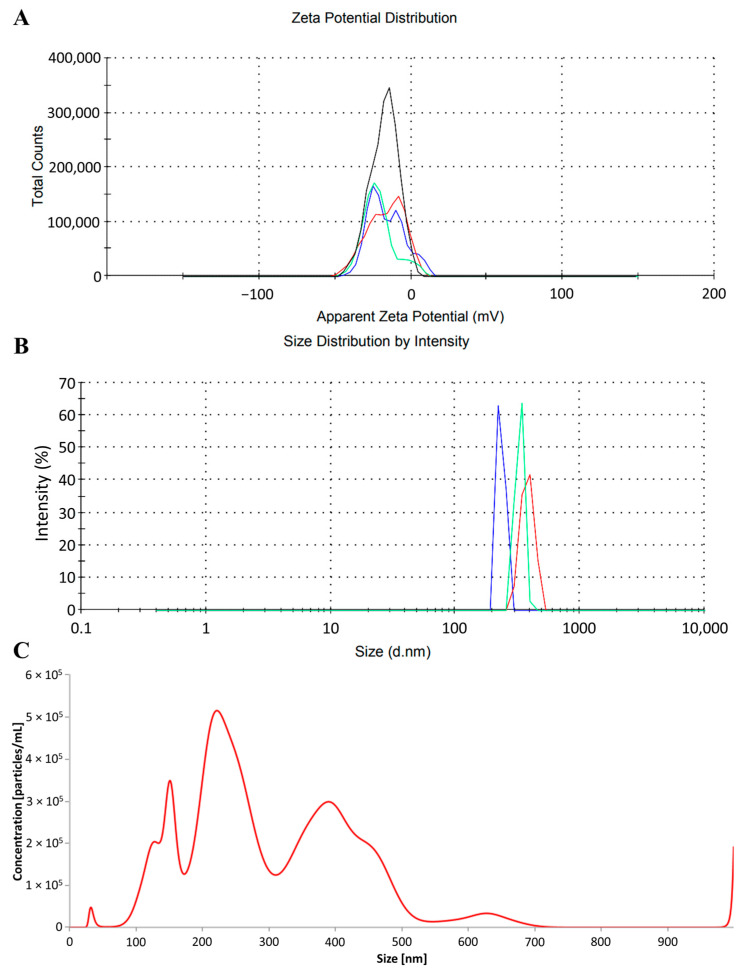
Parameters of the photopolymerizing resin particles (PRPs) utilized in the experiment: (**A**) zeta potential distribution (mV), (**B**) particle diameter distribution measured using dynamic light scattering (DLS) analysis, (**C**) particle diameter distribution measured using nanoparticle tracking analysis (NTA). The blue, green and red lines in panels (**A**,**B**) represent the results of individual measurements. The black line in panel (**A**) indicates the mean result from triplicate.

**Figure 2 ijms-26-11245-f002:**
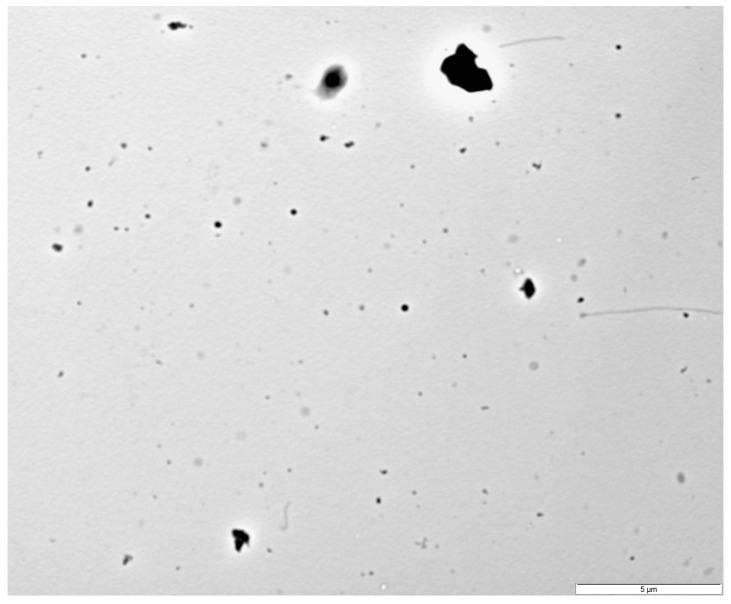
Image of photopolymerizing resin particles (PRPs) and their aggregates using transmission electron microscopy (TEM). Scale bar: 5 µm.

**Figure 3 ijms-26-11245-f003:**
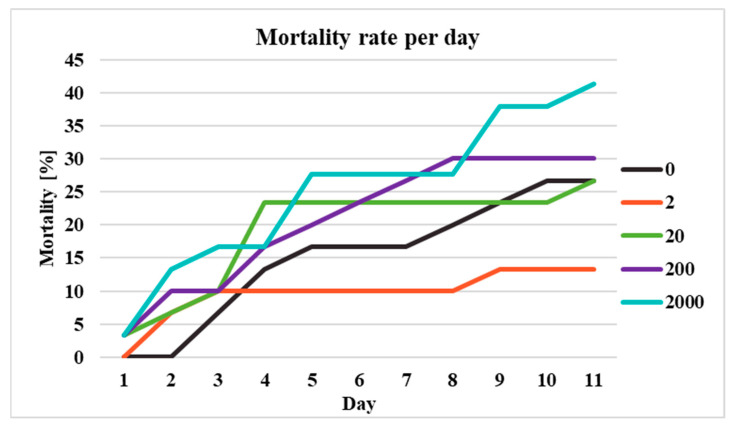
Graph of mortality rate per day. Groups presented as exposure to photopolymerizing resin particles [PRPs] in mg/dm^3^.

**Figure 4 ijms-26-11245-f004:**
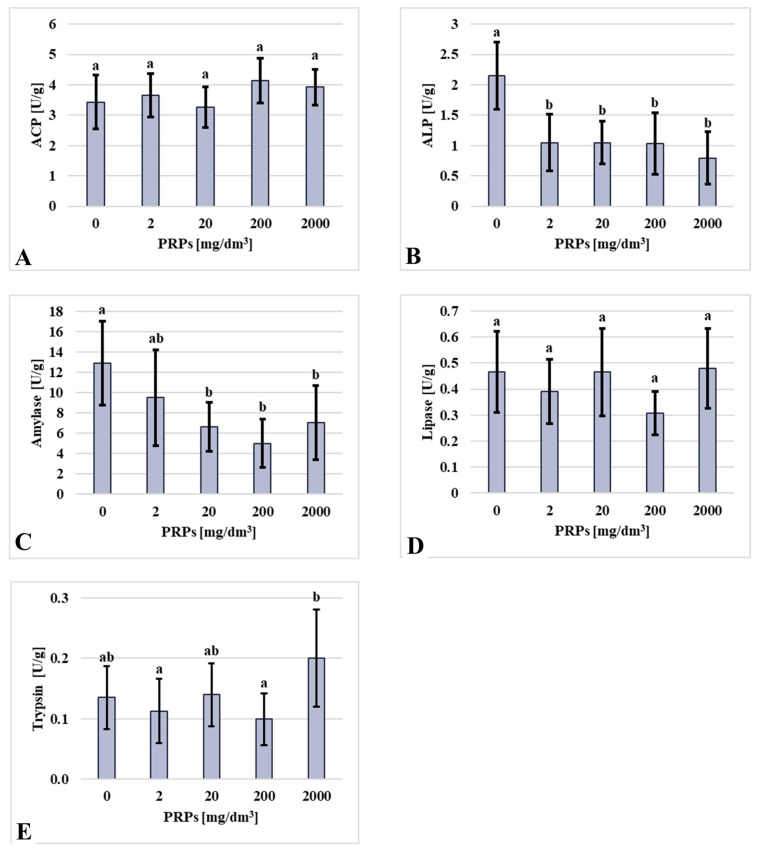
Selected enzyme activities in U/g for proteins (**A**) ACP—acid phosphatase (*n* = 11, 13, 11, 11, 9, respectively, for groups of photopolymerizing resin particles (PRPs): 0, 2, 20, 200, 2000 mg/dm^3^); (**B**) ALP—alkaline phosphatase (*n* = 11, 13, 11, 11, 9, respectively, for groups of PRPs: 0, 2, 20, 200, 2000 mg/dm^3^); (**C**) amylase (*n* = 10, 12, 10, 10, 8, respectively, for groups of PRPs: 0, 2, 20, 200, 2000 mg/dm^3^); (**D**) lipase (*n* = 10, 10, 10, 11, 9, respectively, for groups of PRPs: 0, 2, 20, 200, 2000 mg/dm^3^); and (**E**) trypsin (*n* = 11, 12, 11, 9, 7, respectively, for groups of PRPs: 0, 2, 20, 200, 2000 mg/dm^3^). Values are presented as mean ± standard deviation; lowercase letters indicate statistically significant differences between individual homogeneous groups at *p* < 0.05. The X-axis presents groups as a concentration of PRPs.

**Figure 5 ijms-26-11245-f005:**
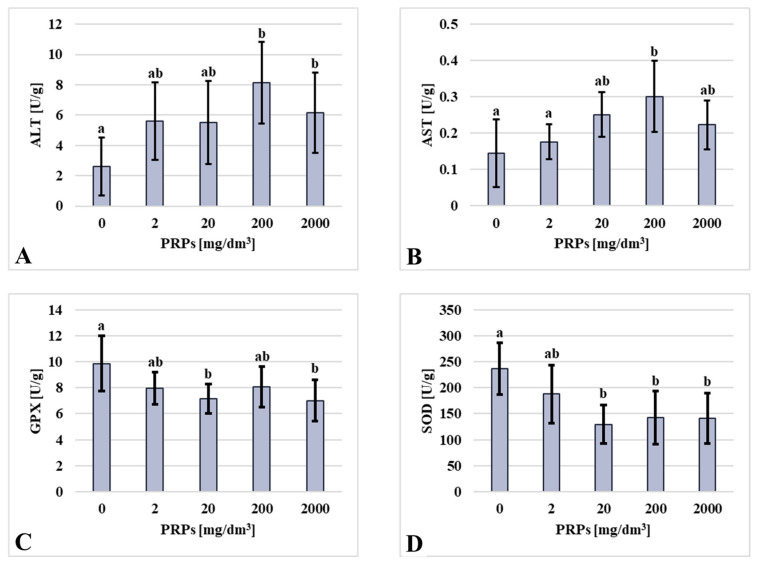
Selected enzyme activities in U/g for proteins (**A**) ALT—alanine aminotransferase (*n* = 11, 13, 11, 11, 9, respectively, for groups of photopolymerizing resin particles (PRPs): 0, 2, 20, 200, 2000 mg/dm^3^); (**B**) AST—aspartate aminotransferase (*n* = 10, 8, 8, 10, 8, respectively, for groups of PRPs: 0, 2, 20, 200, 2000 mg/dm^3^); (**C**) GPX—glutathione peroxidase (*n* = 11, 13, 11, 11, 9, respectively, for groups of PRPs: 0, 2, 20, 200, 2000 mg/dm^3^); and (**D**) SOD—superoxide dismutase (*n* = 11, 13, 11, 11, 9, respectively, for groups of PRPs: 0, 2, 20, 200, 2000 mg/dm^3^). Values are presented as mean ± standard deviation; lowercase letters indicate statistically significant differences between individual homogeneous groups at *p* < 0.05. The X-axis presents groups as a concentration of PRPs.

**Table 1 ijms-26-11245-t001:** Mortality rate across groups at the end of the test (*n* = 30).

Group [mg/dm^3^ of PRP]	Mortality [%]	Mortality Sex Ratio (Male/Female) [%]	Mortality Per Sex (Male/Female) [*n*]
0 (control)	26.67 ± 23.09	42.86:57.14	3:5
2	13.33 ± 15.28	100.00:0.00	4:0
20	26.67 ± 11.55	87.50:12.50	7:1
200	30.00 ± 10.00	88.89:11.11	8:1
2000 *	41.38 ± 10.32	69.23:30.77	9:4

There were no statistically significant differences between the groups for mortality rate. * *n* = 29: see more in Section 4.8 of the Section 4.

## Data Availability

The original contributions presented in this study are included in the article/Appendix A. Further inquiries can be directed to the corresponding author.

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
