# Peer review of "Preliminary Assessment of the Possible Environmental Risks of Photopolymerizing Resin Particles Produced by Finishing Stereolithography 3D-Printed Objects, Employing Toxicity Test on Tropical House Crickets (*Gryllodes sigillatus*)"

_ijms, 2025, doi:10.3390/ijms262311245_

Round 1
Reviewer 1 Report (Previous Reviewer 3)
Comments and Suggestions for Authors
The authors have revised their manuscript and corrected several critical points. Nevertheless, I still have a number of serious questions regarding the methodology and the associated interpretation of the results.
Lines 141-142. 26.67% with a sample size of 30 is 8 dead crickets, not 7 as stated by the authors (3 males and 4 females). Then the ratio between mortality rates for the sexes is not 43:57, as I already mentioned in my previous review. The authors need to clarify this point.
Lines 338-339. In these lines, the authors indicated the gender distribution at the start of the experiment. It follows that there were only 7 males out of 30 individuals in the control group. At the same time, the gender ratio in the experimental groups was close to 50%, or even in some groups, males were numerically dominant, as I pointed out in my previous review. The samples are too different, and based on this, I believe that, despite the interesting theory proposed by the authors, the data obtained do not allow conclusions to be drawn about the increased mortality of males under the influence of the study plastic . I recommend that the authors focus on overall mortality in this manuscript. The data on the relationship between mortality and sex should extend with new experimental work in the future.
Lines 364-366. There is no clear justification for the different samples size that were taken for biochemical studies. Why are they different in different groups? If these are all the surviving crickets, then there should be more of them... I do not understand any other reasons for the different samples. Furthermore, it is unclear why the value of n differs for different enzymes in the captions for figures 4 and 5. The authors need to clarify this points.
There are also complaints about the “conclusion” section.
Lines 466-467. "The dataset indicates that G. sigillatus readily ingests PRPs contaminated agarose gels". The article does not provide any actual data on the absorption of gel by crickets. It only shows the effect of this absorption. The statement should be deleted, rewritten, or confirmed by results.
Lines 467-468. "Toxicity increased in a concentration-dependent manner, as evidenced by elevated mortality rates and alterations in enzymatic activity profiles". In most results, the difference between different concentrations was either not significant at all or was only visible at a concentration of 2000 mg. This is not enough to call the effect dose-dependent. The sentence needs to be rewritten.
Author Response
We would like to express our deepest appreciation for time that reviewer has devoted to the manuscript. Please find below answers to the comments.
Lines 141-142. 26.67% with a sample size of 30 is 8 dead crickets, not 7 as stated by the authors (3 males and 4 females). Then the ratio between mortality rates for the sexes is not 43:57, as I already mentioned in my previous review. The authors need to clarify this point.
We would like to express our deepest gratitude for highlighting this issue. Due to an omission, 4 instead of 5 as the number of decayed females in the control group was entered in the table, which resulted in questionable mortality ratios. All data was checked again to avoid similar mistakes.
Lines 338-339. In these lines, the authors indicated the gender distribution at the start of the experiment. It follows that there were only 7 males out of 30 individuals in the control group. At the same time, the gender ratio in the experimental groups was close to 50%, or even in some groups, males were numerically dominant, as I pointed out in my previous review. The samples are too different, and based on this, I believe that, despite the interesting theory proposed by the authors, the data obtained do not allow conclusions to be drawn about the increased mortality of males under the influence of the study plastic . I recommend that the authors focus on overall mortality in this manuscript. The data on the relationship between mortality and sex should extend with new experimental work in the future.
Sentences to the discussion and conclusion sections were added to emphasize possible interesting findings that should be studied more thoroughly in the future to confirm them with more data.
Lines 364-366. There is no clear justification for the different samples size that were taken for biochemical studies. Why are they different in different groups? If these are all the surviving crickets, then there should be more of them... I do not understand any other reasons for the different samples. Furthermore, it is unclear why the value of n differs for different enzymes in the captions for figures 4 and 5. The authors need to clarify this points.
Half of the surviving crickets were sampled for the biochemical panel; information about the number of sampled crickets is present in the materials and methods. Slight variations in the numbers between various enzymes in the same groups are caused by the removal of outliers. Specified numbers of specimens for each enzyme activity assay per group were added to the figures' captions for further clarification, as well as showed additional omissions which were corrected.
Lines 466-467. "The dataset indicates that G. sigillatus readily ingests PRPs contaminated agarose gels". The article does not provide any actual data on the absorption of gel by crickets. It only shows the effect of this absorption. The statement should be deleted, rewritten, or confirmed by results.
Sentence deleted as suggested.
Lines 467-468. "Toxicity increased in a concentration-dependent manner, as evidenced by elevated mortality rates and alterations in enzymatic activity profiles". In most results, the difference between different concentrations was either not significant at all or was only visible at a concentration of 2000 mg. This is not enough to call the effect dose-dependent. The sentence needs to be rewritten.
The sentence was written again to better emphasize the lack of statistically significant differences in the obtained results.
Reviewer 2 Report (Previous Reviewer 1)
Comments and Suggestions for Authors
This is a resubmitted paper. The author did not respond positively to my comments in the previous submission. I firmly believe that this article lacks novelty, has very little data, and should not be published.
The significance labeling in the author's image is incorrect.
Author Response
We regret that our responses to the previous review did not fully address the reviewer’s concerns and that we were unable to clearly convey the innovative aspects of our study. Nevertheless, we remain sincerely grateful for the reviewer’s thoughtful comments and for the considerable time and effort devoted to evaluating our manuscript.
Reviewer 3 Report (Previous Reviewer 2)
Comments and Suggestions for Authors
This is a comprehensive review of the article "Preliminary assessment of possible environmental risks of photopolymerizing resin particles produced by finishing stereolithography 3D printed objects employing toxicity test on tropical house cricket (Gryllodes sigillatus)." The following remarks are offered for consideration:
- The study used Gryllodes sigillatus as a model organism. While useful for terrestrial toxicity tests, this species does not represent the full diversity of insect life, and other insect species may respond differently to the tested particles.
- The methodology was limited to a 96-hour exposure period, which is considered a short-term exposure and may not accurately reflect long-term environmental effects. Further, the paper itself notes the necessity for longer-term studies to assess chronic effects, as some adverse health effects may only manifest after prolonged exposure.
- The study did not evaluate the impact of the photopolymer resin particles on the terrestrial food chain. Future research should assess how these particles affect the entire food chain.
- The particles used in the study were produced from only one type of resin (Phrozen Aqua-Gray 8K). The authors acknowledged that different resins may behave differently and therefore pose different environmental risks.
- I recommend performing FT-IR spectroscopy on the photopolymerizing resin particles to provide a more detailed chemical analysis.
- Figure 1 (A), (B), and (C) are not of publication quality and should be improved for clarity and presentation.
Author Response
We would like to express gratitude for providing insightful comments of the manuscript. Please find below answers to them.
I recommend performing FT-IR spectroscopy on the photopolymerizing resin particles to provide a more detailed chemical analysis.
Unfortunately, we are unable to perform such an analysis now; however, in the future, with our work with similar xenobiotics, such an analysis will be considered.
Figure 1 (A), (B), and (C) are not of publication quality and should be improved for clarity and presentation.
We have tried further revising graphs; unfortunately, it didn’t help much. Graphs are from software supplied with the nanotracking analysis device and the zetasizer created by Malvern Panalytical Ltd. On further request figure can be split into 3 parts, which might help with the presentation.
Reviewer 4 Report (New Reviewer)
Comments and Suggestions for Authors
The manuscript investigates the preliminary environmental risks of photopolymerizing resin particles (PRPs) produced during stereolithography (SLA) 3D printing, using the tropical house cricket (Gryllodes sigillatus) as a model organism. T
Correct grammatical errors (“This study evaluate” → “This study evaluates”).
The abstract is too detailed in listing enzymes; instead, summarize the main patterns (increase/decrease).
Clarify the novelty of the work by explicitly stating that this is the first study testing polymerized PRPs in G. sigillatus.
The historical background on 3D printing (1970s–1980s) is unnecessarily long; shorten and focus more on environmental risk aspects.
Clearly state the research gap, lack of data on polymerized resin particles in terrestrial models.
End the introduction with precise objectives/hypotheses.
Explain why agarose gels were chosen as a feeding medium and whether PRPs remain evenly distributed or may sediment.
Clarify the rationale for the chosen concentration range; relate it to realistic environmental exposure scenarios.
Apply survival analysis (Kaplan–Meier, Cox regression) rather than only χ².
justify why these particular enzymes were selected and whether they represent validated biomarkers for crickets.
Provide more detail on sample size justification (n per group, statistical power).
Always include the number of replicates (n) in figure captions.
Wide standard deviations in mortality suggest small sample bias—acknowledge this.
Report effect sizes, not just p-values, to assess biological relevance.
Avoid over-speculation about pulverization of particles to nanoparticle scale; this was not directly measured in the study.
Provide alternative explanations for enzyme inhibition/stimulation, such as general stress from diet changes or gel feeding method.
Compare concentrations used with likely environmental exposures; current values (200–2000 mg/dm³) may not be environmentally realistic.
Separate clearly between findings of this study and comparisons with other plastics/nanoparticles; currently they are mixed together and may confuse readers.
Clarify whether sex differences are biologically significant or due to uneven male–female ratios in experimental groups.
In conclusion restates results; instead, highlight the main implication (preliminary evidence of PRP toxicity, need for chronic/multigenerational studies, relevance for waste management policies).
Provide clearer suggestions for next research steps.
Author Response
We would like to express our deepest appreciation for the time devoted for the review of the manuscript and all comments that provided a valuable point of view. Please find below detailed answers to the comments.
Correct grammatical errors (“This study evaluate” → “This study evaluates”).
Grammar has been revised as suggested.
The abstract is too detailed in listing enzymes; instead, summarize the main patterns (increase/decrease).
The list of enzymes has been deleted. Increase and decrease patterns have been added instead as suggested.
Clarify the novelty of the work by explicitly stating that this is the first study testing polymerized PRPs in G. sigillatus.
In both the introduction and abstract sections, statements about the first-ever study on the effects of PRPs on G. sigillatus were added.
The historical background on 3D printing (1970s–1980s) is unnecessarily long; shorten and focus more on environmental risk aspects.
The historical background has been shortened as suggested. Environmental risk aspects were not expanded due to a lack of knowledge, which could be further added.
Clearly state the research gap, lack of data on polymerized resin particles in terrestrial models.
Emphasized that there is a lack of knowledge in terrestrial animals.
End the introduction with precise objectives/hypotheses.
As suggested, a clear objective of the study was added at the end of the introduction.
Explain why agarose gels were chosen as a feeding medium and whether PRPs remain evenly distributed or may sediment.
Various gels, including agarose gels, are commonly used in insect nutrition by hobbyists as well as in toxicological and feeding experiments in insects. Although a slight degree of particle sedimentation may occur despite the rapid polymerization of agarose gel, this method remains one of the convenient and reliable approaches for administering xenobiotics to terrestrial insects. We can be confident that, by consuming the gel, insects ingest the tested substances into their digestive tract, thereby reflecting conditions that can be expected in their natural environment.
Clarify the rationale for the chosen concentration range; relate it to realistic environmental exposure scenarios.
Due to the lack of research focusing on the quantification of PRPs in the environment, the concentration range was chosen based on the most similar xenobiotic, which was studied on the G. sigillatus, and a common practice in toxicological studies that is used for not well-studied xenobiotics has been used.
Apply survival analysis (Kaplan–Meier, Cox regression) rather than only χ².
We thank you for the valuable suggestion. Indeed, the proposed methods would allow for a valuable extension of the presented research. However, as the manuscript reports the results of the first in a series of planned experiments, we intend in the subsequent study—conducted on a larger scale and with a greater number of individuals—to incorporate the reviewer’s suggestion and include the indicated analyses in the statistical evaluation.
justify why these particular enzymes were selected and whether they represent validated biomarkers for various crickets.
Selected enzymes are widely used in toxicology studies; all biomarkers were earlier described and detected in crickets.
Provide more detail on sample size justification (n per group, statistical power).
A sample justification was added in the experiment design section of the materials and methods.
Always include the number of replicates (n) in figure captions.
The number of samples for each assay has been specified in the figure captions.
Wide standard deviations in mortality suggest small sample bias—acknowledge this.
A sentence about standard deviations in mortality has been added as suggested.
Report effect sizes, not just p-values, to assess biological relevance.
We sincerely thank the reviewer for this observation. Pearson correlation analysis had indeed been performed; however, the results were not included in the final manuscript, as not many significant correlations were identified between the measured parameters. We would be glad to provide the results of this analysis upon request.
Avoid over-speculation about pulverization of particles to nanoparticle scale; this was not directly measured in the study.
A clarification about this hypothesis has been added to emphasize that is mechanism should be further studied in order to verify its occurrence.
Provide alternative explanations for enzyme inhibition/stimulation, such as general stress from diet changes or gel feeding method.
As suggested, an alternative explanation has been added.
Compare concentrations used with likely environmental exposures; current values (200–2000 mg/dm³) may not be environmentally realistic.
We agree that the values employed in the study might not be environmentally realistic. Unfortunately, currently comparison of the concentrations utilized in the study is not possible as there is a lack of research on that topic. Due to this reason, in research design we opted for increasing each next concentration 10 times, which is a standard procedure for not well-studied xenobiotics.
Separate clearly between findings of this study and comparisons with other plastics/nanoparticles; currently they are mixed together and may confuse readers.
Grammar has been revised for better differentiation of findings in the discussion.
Clarify whether sex differences are biologically significant or due to uneven male–female ratios in experimental groups.
A sentence about the need for further research on this topic has been added.
In conclusion restates results; instead, highlight the main implication (preliminary evidence of PRP toxicity, need for chronic/multigenerational studies, relevance for waste management policies).
We find the results recap valuable for possible readers, thus decided to leave it in the conclusions section. Topics for future research has been added as suggested.
Provide clearer suggestions for next research steps.
As suggested next research topics were highlighted.
Reviewer 5 Report (New Reviewer)
Comments and Suggestions for Authors
The manuscript presents a novel and well-designed investigation on the environmental dangers of photopolymerizing resin particles using Gryllodes sigillatus as a terrestrial model, and the results shed light on sex-dependent toxicity and enzymatic changes. Minor adjustments are suggested to improve language clarity, figure captions, and terminology, unit, and reference consistency in order to increase readability and effect.

Improve sentence flow, and increase overall intelligibility for an international audience.
Author Response
We would like to express our deepest appreciation for time devoted for manuscript review. Please find detailed answers for comments below.
1- Provide slightly more detail in the Materials and Methods section on particle preparation (e.g., reproducibility of sanding process, particle size distribution consistency).
More details about the sanding process have been added. Particle size distribution was earlier moved to the results section as suggested by another reviewer. Most of the obtained particles were in the range 100-1000 nm, with most of them in the 100-800nm range.
2- Clarify statistical treatment for sex-dependent mortality differences.
Data used for chi chi-square test has been described more.
3- Expand the discussion to better connect findings with recent work on terrestrial ecotoxicology of microplastics and nano plastics, highlighting the novelty of using G. sigillatus as a model.
As suggested additional literature review has been done; unfortunately, no new articles that we were able to access and that would share not only similar mortality findings were found. The introduction part of the discussion has been tweaked to underscore the novelty of G. sigillatus.
4- Ensure all abbreviations are defined at first mention in the main text and abstract.
Usage of abbreviations has been checked and corrected.
5- Improve grammar, sentence structure, and flow in the abstract and discussion (e.g., "This study evaluate" → "This study evaluates").
Grammar has been checked and improved as suggested.
6- Ensure consistent terminology when referring to photopolymerizing resins (PRs) and photopolymerizing resin particles (PRPs).
Terminology of PRs and PRPs has been checked and corrected to remain consistent.
7- Enhance figure captions with clearer explanations of abbreviations (e.g., ALT, AST, GPX, SOD) for readers from outside toxicology.
Figure captions are changed to find the enzyme name more easily.
Improve sentence flow, and increase overall intelligibility for an international audience.
Sentences have been checked and improved.
Round 2
Reviewer 1 Report (Previous Reviewer 3)
Comments and Suggestions for Authors The authors responded to all my comments. I consider the article suitable for publication.Author Response
Dear Reviewer,
Thank you for your review of our revised publication, "Preliminary assessment of the possible environmental risks of photopolymerizing resin particles produced by finishing stereolithography 3D printed objects, employing toxicity tests on tropical house crickets (Gryllodes sigillatus)". We would like to express our gratitude for the time devoted to reviewing the manuscript.
Kind regards,
Authors
Reviewer 2 Report (Previous Reviewer 1)
Comments and Suggestions for Authors
This is a meaningless article, and I have repeatedly rejected it
Author Response
Dear Reviewer,
We regret to hear that your opinion about the article is not changeable. We would like to thank you again for the time devoted to the review process.
Kind regards,
Authors
Reviewer 4 Report (New Reviewer)
Comments and Suggestions for Authors
Add a brief sentence about how the findings may inform future research or environmental policies related to 3D printing waste.
Reference more specific studies that address the environmental concerns of 3D printing, especially around the disposal of photopolymer resins,
Mention the significance of studying G. sigillatus as a model organism earlier to provide more context for your choice.
When discussing mortality data, be sure to clarify what statistical tests were used to determine significance.
Author Response
Dear reviewer,
Thank you for your review of our revised publication, "Preliminary assessment of the possible environmental risks of photopolymerizing resin particles produced by finishing stereolithography 3D printed objects, employing toxicity tests on tropical house crickets (Gryllodes sigillatus)". Please find below detailed answers to your comments.
Add a brief sentence about how the findings may inform future research or environmental policies related to 3D printing waste.
A sentence about possible usage of the findings in both future research and environmental policies has been added to the Introduction. Information about the possible utilization of the results in legislations has also been added to the Conclusion section to complement the presented future research ideas.
Reference more specific studies that address the environmental concerns of 3D printing, especially around the disposal of photopolymer resins.
An additional literature review has been performed, which resulted in the addition of 5 more references connected to photopolymer resin disposal.
Mention the significance of studying G. sigillatus as a model organism earlier to provide more context for your choice.
As suggested, information about research conducted on G. sigillatus has been moved to mention earlier significance of this model species.
When discussing mortality data, be sure to clarify what statistical tests were used to determine significance.
Statistical tests used for each mortality rate data have been specified as suggested.
Kind regards,
Authors

This manuscript is a resubmission of an earlier submission. The following is a list of the peer review reports and author responses from that submission.
Round 1
Reviewer 1 Report
Comments and Suggestions for Authors
The author should add the results of PRPs on other animals in the background.
Why is there only 29 for group-2000 in Table 1.
Why is there such a large difference of n in Figure 2 and 3?
What is the basis for setting the concentration?
This is a very simple article, the author only has data on mortality and enzyme activities, and the n values of these data are inconsistent, which is very confusing.
The entire article lacks sufficient data and innovation, I think it should not be published.
Reviewer 2 Report
Comments and Suggestions for Authors
This paper discusses the assessment of potential environmental risks associated with 3D printing using a toxicity test on the tropical house cockroach (Gryllodes sigillatus). The work in this paper is commendable, but it does not align with the journal's scope. I believe the appropriate journals for this article are public health journals.
Reviewer 3 Report
Comments and Suggestions for Authors
The authors' manuscript addresses the currently important issue of microplastics' interaction with invertebrates (using the house cricket as an example). The authors conducted experimental work and obtained interesting results. However, in its current form, the article raises many questions and cannot be published in IJMS.
Major comments:
Comment 1. Lines 1-3. The title is incorrect—the authors studied the impact of a specific plastic, not the 3D printing process.
Comment 2. Lines 34-64. The introduction is very short; the authors need to expand it.
Comment 3. Based on the “discussion” section written by the authors themselves (Lines 147-176), it can be concluded that, due to the instability of the test object, the mortality results are questionable. However, the main conclusions of the article are based precisely on these results. This is currently the main shortcoming of this work. It raises a number of questions about the methodology and results:
- -The manuscript does not specify whether the sex was determined before the experiment began. It is possible that the different mortality rates among males were caused by their dominance in the selection process. Was the sex ratio in each group 50/50? Were 75 male and 75 female crickets purchased in advance?
- -Both in the results and in the discussion, in addition to mortality, necrophagy, cannibalism, and molting are considered, but nothing is said about them in the materials and methods section.
- -The authors do not explain why mortality is given for the entire sample rather than for individual groups.
- -Why the authors use rounded percentages in the table showing mortality? Moreover, some percentages in the column dedicated to the ratio of mortality by gender are incorrect: for example, if 3 males and 5 females out of 8 died, then the ratio is not 43:57, but 38:62 (37.5:62.5, to be precise) which is not very close to 50:50, as the authors write. These and other values need to be corrected.
Comment 4. Biochemical analysis also raises a number of questions:
- -The results could support the mortality results, but for some reason they are not divided by sex. Why?
- -Virtually none of the enzymes showed a dose-dependent response, even though the working concentrations differed by a factor of 1000. If this is not a methodological error, the authors must to discuss it in the “Discussion” section.
- -Is it appropriate to use whole cricket homogenate for the analysis of digestive enzymes?
Minor comments:
Line 85. Use black for the font in Figure 1.
Lines 130-146. This introductory part of the discussion is too lengthy and unrelated to the results; it should either be shortened or partially moved to the “introduction” section.
Lines 199-209. Why do the authors use data on the toxicity of inorganic nanoparticles in their discussion? Is the authors' main idea that the toxicity of the selected plastic on crickets is related only to the physical parameters of the plastic itself? Does the chemical composition not matter?
Lines 243-270. The results of the particle size analysis should be presented in the “results” section.
Lines 272. It is difficult to judge the morphology of particles based on this figure, which is not done. The name of the figure should be changed.
Lines 275-277. The figure is captioned as a description of a silver nanoparticles. Either the figure or the caption is incorrect. Correct this.
I hope my comments will help you improve your manuscript.